

# A Novel Framework for Calibration and Evaluation of Hydrological Models in Dynamic Catchments

Tian Lan[1,2,3], Xiao Wang[4*], Hongbo Zhang[1,2,3], Xinghui Gong[1,2,3], Xue Xie[5], Yongqin David Chen[6], Chong-Yu Xu[7]

[1]School of Water and Environment, Chang'an University, Xi'an 710054, China.

[2]Key Laboratory of Subsurface Hydrology and Ecological Effects in Arid Region of the Ministry of Education, Chang'an University, Xi'an 710054, China.

[3]Key Laboratory of Eco-Hydrology and Water Security in Arid and Semi-Arid Regions of Ministry of Water Resources, Chang'an University, Xi'an 710054, China.

[4]State Key Laboratory of Water Resources Engineering and Management, Wuhan University, Wuhan, China

[5]Center for Water Resources and Environment, and Guangdong Key Laboratory of Marine Civil Engineering, School of Civil Engineering, Sun Yat-sen University, Guangzhou 510275, China

[6]School of Humanities and Social Science, The Chinese University of Hong Kong, Shenzhen 518172, China.

[7]Department of Geosciences, University of Oslo, P.O. Box 1047 Blindern, 0316 Oslo, Norway.

*Correspondence to*: Xiao Wang (xiao_wang@whu.edu.cn)



**Abstract**

Hydrological models often face challenges in accurately simulating dynamic catchment processes due to structural deficiencies caused by oversimplifications. This results in compromised accuracy in capturing dynamic behaviours across different flow phases in seasonal catchments. To address this challenge, this study proposes a robust calibration framework that incorporates dynamic

catchment characteristics. Additionally, the potential impacts of objective function configuration and sub-period calibration with dynamic parameters were investigated in this study. A pre-processing framework was developed to bridge models with catchment dynamics by clustering time series into sub-periods with similar hydrological processes. Seven calibration experiments were conducted to explore issues related to time-invariant parameters, objective function configurations, parameter correlations, dimensionality in global optimization, and abrupt parameter shifts. The experiments were conducted using the MOPEX dataset,

which includes 219 basins across the United States, and were evaluated based on performance metrics, as well as state variables and fluxes. The recommended calibration scheme effectively addressed challenges in dynamic parameter operations, significantly improving model performance across different flow phases and enhancing the simulation in dynamic catchments. In conclusion, incorporating dynamic parameters based on extracted catchment characteristics effectively mitigates structural deficiencies in hydrological models. This approach improves simulation accuracy across different flow phases, reduces uncertainty, and enhances

the model's ability to capture dynamic hydrological processes in seasonal catchments. Our findings provide a practical solution for calibrating hydrological models in seasonal catchments, contributing to better understanding of the hydrological cycle.

**1 Introduction**

Hydrological modelling is a crucial tool for water management, facilitating the management of hydrological challenges such as runoff prediction, disaster warning, and water resource management (Gosling et al., 2011; Shrestha et al., 2021; Grayson et al.,

2010). However, understanding, modelling, and predicting hydrological processes with greater realism remains significant challenges in hydrological sciences (Clark et al., 2016). In many cases, when hydrological observational data are limited, process-driven hydrological models are more reliable than data-driven models for understanding catchment hydrological processes (Lakshmi and Sudheer, 2021; Bárdossy, 2007). Although more complex models, such as data-driven deep learning models, which possess several orders of magnitude more parameters (Kratzert et al., 2019a; Nearing et al., 2021), can model highly non-linear

systems and often provide more accurate streamflow predictions compared to process-based models (Kratzert et al., 2018; Kratzert et al., 2019a; Kratzert et al., 2019b; Nearing et al., 2021), the "black box" nature of purely data-driven models prevents a deeper understanding of hydrological processes (Karpatne et al., 2017). One of the most common flaws in the structure of process-driven models is their inherent oversimplification, leading to an incomplete representation of dynamic features driving the catchment (Pathiraja et al., 2016; Deng et al., 2016; Wang et al., 2022b). These simplifications, originating from a limited understanding of

the physical mechanisms behind dynamic catchment characteristics (e.g., seasonal changes in climate and land cover), often lead to structural deficiency and compromised model accuracy (Xiong et al., 2019b; Pathiraja et al., 2016; Fowler et al., 2022). The oversimplification issue further manifests in the practice of using time-invariant model parameters (Zhou et al., 2022; Shamir et al., 2005), which focuses on the overall performance of the model, averaging the hydrological responses, resulting in sacrificing the simulation accuracy of high (low) flow to improve the simulation accuracy of low (high) flow (De Vos et al., 2010; Lin et al.,

2010). To address this issue, it is imperative to re-examine historical hydrological and meteorological data, extract catchment dynamics and address structural deficiencies in the models.





Traditional calibration of hydrological models typically employs global evaluation metrics and time-constant parameters, focusing on the model's overall performance. However, this approach might average hydrological responses and fail to ensure accurate simulations across various flow phases and observational periods. In critical runoff events like floods and droughts, this static approach may fail to capture the dynamic nature of hydrological processes, underscoring the need for more flexible calibration methods (Pfannerstill et al., 2014; Clark et al., 2021). Hence, researchers have investigated various calibration techniques incorporating dynamic catchment characteristics. One method involves revising the objective function based on selected evaluation criteria to improve model performance (Thyer et al., 2009; Ji et al., 2023). Calibrations using multi-objective optimization algorithms better highlight different flow phases, but face potential challenges such as increased computational complexity, sensitivity to parameter settings, and slower convergence with more objective functions (Yapo et al., 1998; Shafii and De Smedt, 2009). Alternative approaches, like multi-weighted objective functions, can improve the simulation accuracy of specific time and flow phases. While these methods enhanced different flow phases and water balance, they may not effectively address structural deficiencies and cannot fundamentally enhance the model's overall performance (Kollat et al., 2012; Fowler et al., 2018; Wagener et al., 2003).

Another strategy involves using dynamic parameters in hydrological models. Implementing dynamic parameters can address limitations in model structure and fundamentally improve predictive performance, not only for specific flow phases or periods but across the entire spectrum of hydrological processes (Zhang and Liu, 2021; Krapu and Borsuk, 2022). Recent studies have significantly advanced hydrological simulations by integrating the dynamic characteristics of catchments. Clustering based on catchment characteristics, such as precipitation, evapotranspiration, and soil moisture, facilitates the segmentation of dynamic hydrological processes into distinct sub-periods (Choi and Beven, 2007; De Vos et al., 2010; Lakshmi and Sudheer, 2021). Wei et al. (2021) further broadened this perspective by highlighting the hydrological processes that arise from the interplay of various factors, including meteorological conditions, surface characteristics, and anthropogenic interference. This interaction among water balance components, such as soil, vegetation, and topography, exhibits temporal variability, which ideally should be captured by process-driven hydrologic simulation models. These changes need to be taken into account through model parameters (Bronstert, 2004; Hundecha and Bárdossy, 2004). Zhang and Liu (2021) suggested that temporal variations in parameters reflect the evolving environment. However, some fundamental problems still need to be addressed before applying the dynamic parameters. Sub-period calibration with dynamic parameters involves the hydrological model structure, global optimization, physical mechanisms of dynamic catchment characteristics, as well as complex relationships between the parameters, state variables and fluxes.

This study aims to investigate the challenges encountered during calibration operations in hydrological models, particularly in catchments with seasonal dynamics, and propose a robust calibration scheme to address these issues. The research focuses on two primary areas: the impact of objective function settings and the complications arising from sub-period calibration with dynamic parameters. To address these concerns, a series of experiments are conducted. First, experiments 1 to 3 examine whether adjusting objective functions can enhance model performance across different flow phases, with a traditional objective function serving as the control (Kollat et al., 2012; Fowler et al., 2018; Wagener et al., 2003). Second, experiments 4 to 6 explore the effects of parameter correlation (Wagener and Kollat, 2007; Höge et al., 2018; Bárdossy and Singh, 2008), the dimensionality disaster on global optimization (Orth et al., 2015; Daggupati et al., 2015; Xie et al., 2021), and the uncertainty introduced by abrupt parameter shifts (Kim and Han, 2017). The performance of these experiments is evaluated using multi-performance metrics, with model state variables and fluxes providing insights into the model's internal behaviours. Ultimately, experiment 7 provides a recommended optimal scheme for modelling seasonal catchments, offering a balanced solution to the identified calibration challenges. All





experiments are verified using the MOPEX dataset, and the recommended calibration scheme is demonstrated through four case
studies, illustrating its improvements in addressing the identified calibration challenges.

## 2 Dynamic catchment characteristics

### 2.1 Study area

The Model Parameter Estimation Experiment (MOPEX) is an international project aimed at developing enhanced techniques for
the priori estimation of parameters in hydrologic models and land surface parameterization schemes of weather and climate models
(Duan et al., 2006). A comprehensive MOPEX database has been developed that contains historical hydrometeorological data and
land-surface characteristics data for numerous hydrologic basins in the United States (US) and other countries. This study utilizes
the dataset from 219 basins spatially distributed across the contiguous US (Fig. 1(a)). Rigorous screening criteria were applied to
ensure the acquisition of high-quality data. The screening process involved three key considerations: (1) no missing or abnormal
data throughout the study period; (2) minimal interference from anthropogenic influences in both temporal and spatial dimensions;
and (3) large spatial distribution scale of the selected basins, including diverse meteorological and underlying surface conditions.
The dataset for selected basins includes the hydrometeorological forcing data, land-surface data, and streamflow data, covering the
period from 1983 to 2000. Hydrometeorological data includes daily precipitation data ($P$), temperature data ($T$), and streamflow
($Q$) provided by the MOPEX dataset, as well as potential evaporation data ($PE$) calculated by the Hamon model (refer to Supporting
Information S2.1) (Mccabe et al., 2015). The Normalized Difference Vegetation Index (NDVI) was used as one of the land-surface
indices to represent the vegetation coverage of the basins, which had a spatial resolution of 8 km and a temporal resolution of half-
monthly intervals (Tucker et al., 2010). Based on these criteria, a total of 219 catchments were selected (Fig. 1(a)), spanning a
wide range of hydrologic and meteorological characteristics, making them ideal for testing various model structures under diverse
conditions (Duan et al., 2006). Additionally, four catchments—Case A (N13302500), Case B (N04073500), Case C (N06192500),
and Case D (N08085500)—are analysed in more detail as case studies.

### 2.2 Catchment dynamics

A systematic approach for Extracting Seasonal Dynamic Catchment Characteristics (EDCC) has been developed. This method
utilizes machine learning techniques to extract seasonal dynamic information from the catchment, and finally divide time series
data into periods with similar seasonal dynamics for further analysis. This process includes three main steps: ① Data Sampling: A
15-day moving window is used to sample hydro-meteorological data, capturing early-stage catchment characteristics. A seasonal
characteristic index system is developed, including climatic indices (e.g., precipitation, temperature, evaporation) and land-surface
indices (e.g., NDVI, antecedent soil moisture). ② Data Cleaning: A Seasonality Index ($SI$) is calculated to analyse temporal
patterns and identify significant seasonality. The Maximal Information Coefficient (MIC) is used to select indices significantly
correlated with streamflow, and Principal Components Analysis (PCA) is applied to eliminate redundant information. ③ Data
Clustering: The Fuzzy C-Means (FCM) algorithm is used to cluster data based on climatic and land-surface indices.

The sub-period division results for the four study cases are illustrated in the hydrograph presented in Fig. 1(c), where identical
colours denote sub-periods with similar catchment dynamics. A detailed description of the EDCC approach is provided in
Supporting Information S2.



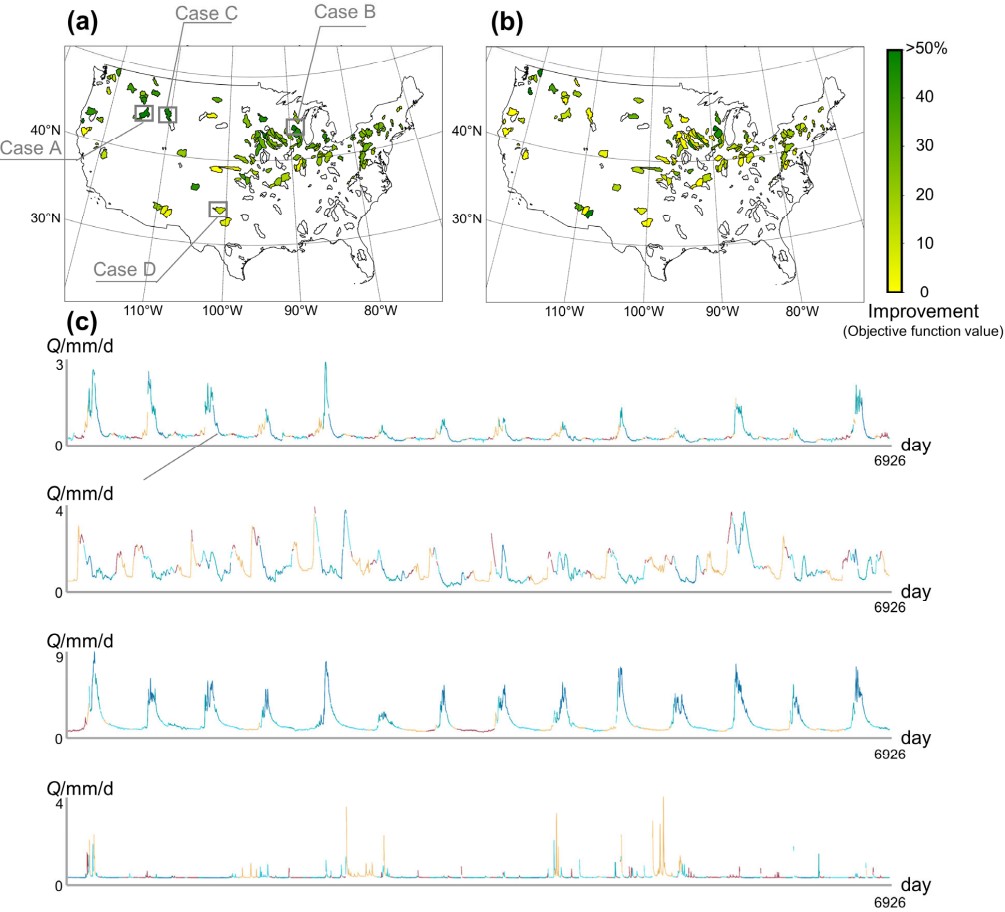

**Figure 1**. **(a),** Location map of the catchment area used in this study, where cases A, B, C, and D correspond to basins N13302500, N04073500, N06192500, and N08085500, respectively. The shading indicates the improvement in the objective function during the calibration period for the recommended scheme, as detailed in Section 4.1. **(b),** Improvement during the validation period, with shading reflecting results from Section 4.1. **(c),** Visualization of clustering results on the hydrograph for the respective study cases.

## 3 Methods

### 3.1 Calibration

Seven calibration experiments are designed and compared to explore potential issues in model calibration for seasonal catchments, and a recommended scheme is proposed based on the results (see Fig. 2). These experiments assess challenges associated with calibration dynamic catchments. Experiments 1 to 3 compare the conventional calibration experiment, the Pareto-based multi-objective calibration experiment, and the multi-weighted objective functions calibration experiment. These experiments are employed to investigate issues related to time-constant parameters, the trade-off between objective functions, and the emphasis across different flow phases. Additionally, three sub-period calibration experiments (experiments 4-6) are utilized to explore





problems brought by dynamic parameters, including correlation between parameters, the dimensional disaster of parameters, the abrupt shift of parameters and the transition of state variables and fluxes between sub-periods.

### 3.1.1 Hydrological model

For illustrative purposes, the HYMOD (Hydrological MODel) model (Moore, 2009) is utilized in this study, since HYMOD is a conceptual rainfall-runoff model with a simple structure (five parameters), low input requirements, and clear physical meanings (Fig. 2). The HYMOD consists of a soil-moisture-accounting module with three adjustable parameters: $H_{uz}$, $B$, and $\alpha$, and two flow-routing modules with two adjustable parameters: $K_q$ and $K_s$. The definition of the model parameters, state variables, and fluxes are presented in the Supporting Information (Table S1). To transform snowfall into effective precipitation, the Degree-day model,

a widely employed method in hydrology and climatology for estimating snowmelt in snowy basins or glacierized areas based on temperature data is applied in this study (see Supporting Information S2.6) (Wang et al., 2022a).

  Throughout the experiments, the Shuffled Complex Evolution algorithm (SCE-UA) is employed to search for the globally optimal parameter set (see Supporting Information S2.5) (Duan et al., 1993). All parameters not mentioned are set to their default values. Considering both high flow and low flow, the objective function is set as $OF = 1 - 0.5 \cdot (\text{NSE} + \text{LNSE})$, where NSE is sensitive

to peaks and discharge dynamics, while the LNSE emphasizes low flows due to the logarithm of discharge (Nash and Sutcliffe, 1970) The HYMOD model was configured for basins over a 19-year period from 1982 to 2000, with the first year serving as warm-up period, the following 13 years serving as the calibration period and the last 5 years as the validation period. It should be noted that this framework is applicable to all models, and the use of the HYMOD model is merely for the purpose of clearly demonstrating the dynamic changes in parameters and model operations.

### 3.1.2 Calibration experiments


  **Experiment 1**, as a control scheme, is employed to compare and investigate potential issues arising from time-invariant parameters. The global optimization procedure with time-constant parameters focuses on the overall performance of the hydrological model. However, it may average the hydrological responses and fail to ensure accurate simulation across all time intervals or flow phases. Fundamentally, the use of time-constant parameters may result in model simulations that compromise accuracy during high-flow

phases while improving accuracy for low-flow phases. This implies a trade-off where the pursuit of enhanced accuracy in low-flow simulations comes at the expense of accuracy in high-flow simulations, and vice versa (Xiong et al., 2019a; Deng et al., 2016; Wang et al., 2017).

  **Experiment 2** investigates solving complex optimization problems with multi-objective optimization. The Non-dominated Sorting Genetic Algorithm II (NSGA-II) is selected for achieving global parameter optimization. In hydrological models, NSGA-II

typically configures a pair of opposing objective functions, such as performance metrics for high and low flows (e.g., NSE and Log NSE). However, the dynamic characteristics of the extracted catchment often exhibit greater complexity, which may not be fully captured by the objective functions used in NSGA-II. It is important to note that, despite the continuous advancements and refinements in Pareto-based multi-objective optimization algorithms, challenges such as heightened computational complexity, sensitivity to parameter settings, and potential slower convergence persist, especially with an increasing number of objective

functions. Furthermore, while a diverse set of optimal parameter sets along the Pareto front improves low-flow simulation accuracy at the expense of high-flow simulation accuracy, it does not inherently address model structural deficiencies or enhance the overall performance of hydrological models (Kollat et al., 2012; Fowler et al., 2018; Wagener et al., 2003).





**Experiment 3** is dedicated to investigating the impact of objective function configuration on model performance. The objective function is designed to encompass diverse flow phases ensuring simulation accuracy across various conditions. This is crucial for

risk management of flood and drought events. To address this, a multi-weighted objective function is proposed, integrating both the RMSE and the FDC. RMSE is advantageous due to its sensitivity to prediction errors, while FDC serves as a hydrological tool graphically illustrating the proportion of time various flow rates are equalled or exceeded. The simultaneous incorporation of FDC and RMSE, specifically calculated the RMSE_Q95, RMSE_Q70, RMSE_Qmid, RMSE_Q20, and RMSE_Q5 based on no exceedance probability, addresses the simulation accuracy of extremely high, high, moderate, low, and extremely low flows,

respectively (Pfannerstill et al., 2014). The weights of each part of the objective function are derived from the results of experiment 1, where the weights were assigned using a combination of AHP, PP, and CRITIC methods to evaluate the results of each part of the objective function. For detailed information, please refer to Supporting Information S2.7. This approach allows decision-makers to assign weights to different flow phases based on practical considerations while ensuring an objective and scientific evaluation.

**Experiment 4** is a sub-period calibration scheme specifically designed to investigate the impact of complex correlations among parameters. Parameter with highest sensitivity is enabled to change over time to bridge the gap between the extracted dynamic information from EDCC and the hydrological model. When considering the dynamization of individual parameters based on the extracted information, the correlations among parameters may lead to dynamic changes in one parameter being counteracted by adjustments in other fixed parameters, which is also called "compensation" (Wagener and Kollat, 2007). As a result, the extracted

dynamic catchment characteristics may not be effectively reflected in the hydrological model by making one parameter dynamic, resulting in limited improvements to the model's accuracy. In this regard, this experiment designated the parameter with the highest sensitivity as the dynamic parameter.

Variance Based Sensitivity Analysis (VBSA) is employed to perform global sensitivity analysis and to quantify the individual contributions of each parameter to the overall variability in the model output (Pianosi et al., 2015). During the calibration period,

the dynamic parameter and other fixed parameters are simultaneously optimized. For example, in the HYMOD model with five parameters, if $H_{uz}$ (maximum height of the soil moisture accounting tank) shows the highest sensitivity and there are five sub-periods, a total of nine parameters are optimized in each model run—$H_{uz}$ and the four fixed parameters across the five sub-periods. The transition of state variables and fluxes between two consecutive sub-periods is achieved by inheriting the last values of the former period as the initial values of the next period. In the validation period, the model is run with the specified dynamic parameter

and other fixed parameters. The transitions of parameters, state variables, and fluxes between two consecutive sub-periods are handled similarly to the calibration period.

**Experiment 5** is dedicated to investigating the potential challenges arising from the dimensional disaster of parameters. The determination of the number of sub-periods is influenced by both the dynamic characteristics of the catchment and the unsupervised clustering of hydrological processes. As the number of sub-periods increases, the number of parameters in sub-period calibration

grows multiplicatively. For example, in the study case area with $n$ sub-periods ($n$ is the number of the sub-periods), the number of parameters involved in the HYMOD model calibration is $5 \times n$. The simultaneous operation of high-dimensional parameters poses the risk of triggering the curse of dimensionality for hydrological model parameters, possibly resulting in the instability of the optimization algorithm and hydrological model. To address the issues associated with high-dimensional parameters in sub-period calibration, this study examines the simultaneous optimization with an increasing multiplicative number of parameters, with the

transition of state variables and fluxes between adjacent sub-periods following the same principles as experiment 4.



**Experiment 6** investigates the potential issues arising from the abrupt shift of dynamic parameters between sub-periods. During the calibration period, the model runs over the whole period, with the optimal performance in each sub-period serving as the objective function for each run. For instance, in a basin divided into five distinct sub-periods, the model runs $n$ iterations ($n$ is the number of the sub-periods), optimizing parameters for the specific objective function corresponding to each sub-period during

each run. Subsequently, the simulated flow data from each sub-period are then merged and compared with the observed flow. In the validation period, the transitions of parameters, state variables, and fluxes between two consecutive sub-periods are handled the same as in experiment 5. The model runs one time using the dynamic parameter sets and the parameter set is switched between two consecutive sub-periods. As a result, the transition of the state variables and fluxes between two consecutive sub-periods is abrupt and achieved by inheriting the last values of the former period as the initial values of the next period. Experiment 6 is

designed with the purpose of mitigating the effects of the correlations and high dimensions of the parameters are excluded, and the influences caused by the parameter transitions are elucidated (Kim and Han, 2016).

**Experiment 7** is a recommend solution aimed at addressing the aforementioned potential issues. In the calibration period, the model run follows the same procedure as that of the calibration period of experiment 6. In the validation period, the simulated flow data from each sub-period are merged and compared with the observed flow. Specifically, the model runs $n$ times, combining the

simulated flow data in the sub-periods.

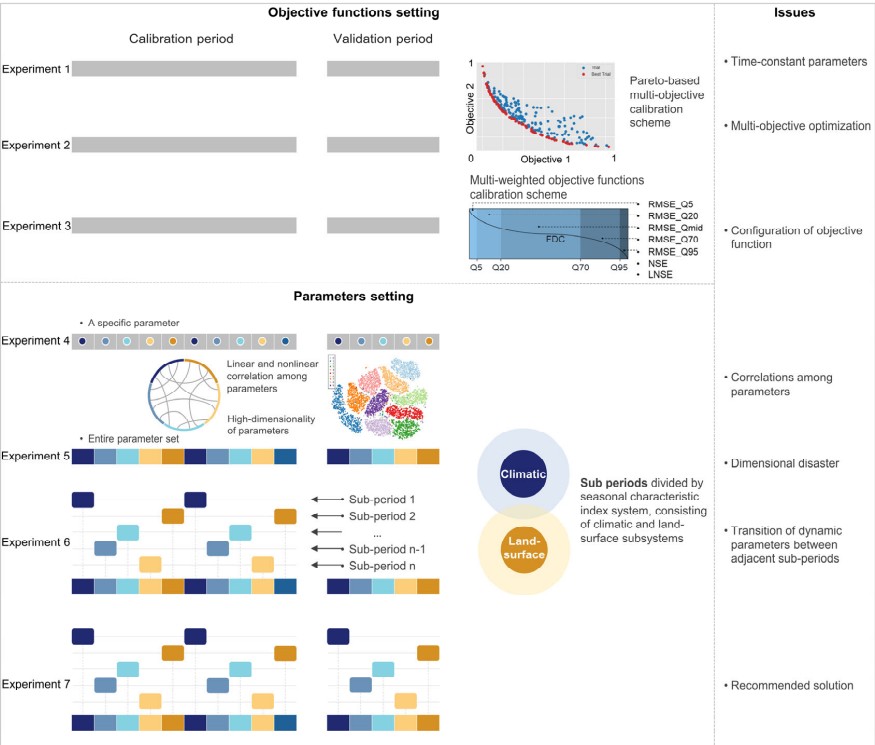

**Figure 2.** Schematic illustration of the seven calibration experiments. The colour bands represent state variables and fluxes, which are continuously transferred within the same period. In experiments 1, 2, and 3, the parameters are time-invariant, but experiments differ in their objective function configurations. Conversely, in experiments 4, 5, and 6 maintain a consistent objective function, but vary the parameters across

different experiments. In experiment 4, the dynamic of only the specific parameter is operated, and the other fixed parameters are optimized





simultaneously. in experiment 5, the parameter set is dynamized. The parameter sets in different sub-periods are optimized simultaneously. In experiment 6, the data from the individual sub-periods are used for minimizing the objective function, while the model is run for the whole period. In the validation period, the parameter set between two consecutive sub-periods is updated accordingly. In experiment 7, the calibration is the same as in experiment 6. In the validation period, the simulated flow data from each separate sub-period are combined and compared with the observed flow.

### 3.2 Evaluation

#### 3.2.1 Multi-criteria evaluation

Model simulations are typically evaluated using performance metrics, which can be divided into statistical and signature metrics (Pfannerstill et al., 2014; Yilmaz et al., 2008; Van Werkhoven et al., 2009). However, a limitation exists with many common performance metrics: they only focus on overall or specific segments of the discharge series, neglecting other parts that may have the greatest practical impact. Hence, for diagnostic analysis, discharge segments of the flow duration curve (FDC) are used to identify discharge levels where model performance is poor (Pfannerstill et al., 2014; Laaha and Blöschl, 2006; Smakhtin, 2001). In this study, performance across different dimensions of streamflow is assessed using the criteria defined in Table 1, which provides a comprehensive evaluation of model performance.

**Table 1.** Description of performance metrics.

| Metric | Description |
|---|---|
| NSE | Sensitive to peaks and discharge dynamic |
| LNSE | Emphasizing low flows with log of discharge |
| RMSE_Q5 | RMSE in FDC Q5 very-high-segment volume |
| RMSE_Q20 | RMSE in FDC between Q5 and Q20 high-segment volume |
| RMSE_Qmid | RMSE in FDC between Q20 and Q70 mid-segment volume |
| RMSE_Q70 | RMSE in FDC between Q70 and Q95 low-segment volume |
| RMSE_Q95 | RMSE in FDC Q95 very-low-segment volume |
| RMSE | RMSE sensitive to flood peaks |
| MSE | MSE sensitive to high flow |
| MSEL | MSEL sensitive to low flow |
| MAE | MAE measuring the overall discharge |

Note that the flow duration curve (FDC) is usually split into different segments to describe different flow characteristics of a catchment (Gupta et al., 2009b; Cheng et al., 2012; Pfannerstill et al., 2014). The RMSE with quadratic character is usually used to evaluate poor model performance due to the strong sensitivity to extreme positive and negative error values.

#### 3.2.2 State variables and fluxes

The internal behaviour of the hydrological model, involving the time series of state variables and fluxes that constitute subspaces within the model space, is visualized in graphs and categorized by the operation of different sub-periods. This visualization helps illustrate issues with calibration experiments. For instance, unreasonable values exceeding operational boundaries often signal errors in model operation triggered by abrupt parameter shifts. Similarly, unresponsive values may indicate either operational errors or unique catchment characteristics. Furthermore, flux map is developed and applied to evaluate the equifinality or uncertainty of internal model behaviour by plotting different components of model fluxes (Khatami et al., 2019). The flux map is a ternary or binary plot where each dimension represents a model runoff flux, and each model run is projected as a single point based on the proportions of its equifinal runoff fluxes to the total simulated $Q$. To HYMOD model, the components with $Q_{q1}$, $Q_{q2}$, and $Q_s$ were defined, which represents the runoff component of the output of quick-release reservoirs of linear routing component





(*OV*₁), the output of quick-release reservoirs of nonlinear routing component ($OV_2$) and the output of slow-release reservoir ($Q_s$).

The point cloud pattern from ternary or binary plots can vary from very constrained up to filling the entire plausible flux space, which represents the different dominant components of runoff. Thus, the point cloud on the flux maps is an expression of the model uncertainty; filling a larger space on the flux map indicates higher degrees of model uncertainty.

## 4 Results

### 4.1 Model performance

Figure 5(a) presents the model performance of the seven experiments in four study cases. In case A, experiment 1 exhibited poor simulation accuracy and weak parameter transferability in diverse flow phases, particularly in extremely low and high flow phases. Experiment 2 demonstrates improved performance in its targeted flow phases. However, this advantage comes at the expense of sacrificing accuracy in other flow regimes. For instance, in experiment 2-1, when focusing on the NSE, case A experienced a decrease in NSE from 0.40 to 0.34 during the calibration period, and from 0.54 to 0.29 in the validation period. However, the

LNSE increased from 0.61 to 2.02 during the calibration period, and from 0.44 to 1.27 in the validation period. When focusing on balancing NSE and LNSE, the parameter sets on the Pareto optimal frontier are similar in performance to those obtained in experiment 1, with NSE and LNSE during calibration being 0.48 and 0.64, respectively, and during validation being 0.48 and 0.44. The results in experiment 2 show limitations on both objective functions compared to the recommended scheme (experiment 7), which demonstrates superior results: NSE and LNSE during calibration are 0.19 and 0.28, respectively, and during validation are

0.21 and 0.24. Despite prioritizing high and low flows through a weighted objective function ($OF = 0.27 \cdot \text{RMSE\_Q5} + 0.16 \cdot \text{RMSE\_Q20} + 0.08 \cdot \text{RMSE\_Qmid} + 0.24 \cdot \text{RMSE\_Q70} + 0.25 \cdot \text{RMSE\_Q95}$ ), experiment 3 underperforms compared to experiment 1. While the objective function emphasizes these targeted phases, adjusting its weights unexpectedly failed to improve performance in the target flow phase and even worsened the model's performance in other evaluation metrics, indicating that this scheme exhibits instability in its performance across different flow phases. Experiment 4 demonstrates marginal improvements

compared to experiment 1 across most metrics. Experiment 5 exhibits the poorest overall model performance during both calibration and validation periods. In some basins, experiment 5 led to operational errors, resulting in invalid results. Experiments 6 and 7, utilizing the same calibration procedures, achieved the best overall performance during the calibration period, particularly in high flow and flood peak responses. However, in the validation period, experiment 6 displayed inconsistent performance, excelling in certain metrics, such as high flow, but showed significant deterioration in others (NSE, LNSE, and RMSE_Q5). Similar

to experiment 5, it may yield invalid values due to the failure of the hydrological model triggered by the calibration scheme. Notably, experiment 7 maintained excellent performance in the validation period, mirroring its calibration results and surpassing other experiments in nearly all metrics. Furthermore, analysis of parameter transferability revealed the smallest differences between calibration and validation periods for experiment 7, while experiment 6 exhibited the largest discrepancies. Hence, experiment 7 demonstrates the superior performer across all evaluation metrics, exhibiting improvements in simulations across various flow

phases. In cases B, C, and D, the evaluation results are similar to those derived from case A.

To further expand the scope of testing to 130 basins in the MOPEX dataset, which exhibit significant seasonal dynamics (refer to Fig. 1), experiment 7 outperformed control experiment 1, demonstrating significant improvement in simulation performance. The evaluation metrics enhanced from 0.43 to 0.34 during the calibration period and from 0.52 to 0.44 during the validation period. In summary, experiments 2 and 3 exhibit performances comparable to experiment 1. Experiment 4 displays slightly superior results.





Experiments 5 and 6 manifest poor overall performance, raising the possibility of generating erroneous outputs. Experiment 7 emerges as the best performer across all evaluation metrics for both calibration and validation periods, demonstrating consistent improvements in hydrological simulations across diverse flow phases.

**4.2 State variables and fluxes**

The state variables and fluxes reflect the internal operation of the hydrological model (definitions are provided in Table S1 in the
Supporting Information). The assessment results of state variables and fluxes through seven calibration experiments for case study A are illustrated in Fig. 3 and Fig. 4 (results of cases B, C and D are shown in S5 of Supporting Information). Experiments 1, 2, and 3 exhibited only minimal differences in both state variables and flux time series, with only the results of experiments 1 and 3 shown for brevity. Experiment 4, contrary to expectations, did not improve the model's performance. Notably, the state variable $X_q$ and flux $Q_q$ in experiment 4 display abnormally flat compared to the control experiment, indicating a wrong response of the
rapid runoff module to input variations. Experiment 5 exhibits invalid values in both state variables and flux time series, due to abrupt changes at the transitions between sub-periods. Experiment 6 in the validation period exhibits similarities to experiment 5.

Despite these setbacks, experiment 7 introduced significant improvements, marking a turning point in the simulation results. In experiment 7, flux responses differ from those in experiment 1 as follows: (1) Sub-periods 1 and 2 in case A, identified by the chosen seasonal characteristics, are the coldest with the least runoff. Although significant one-day maximum precipitation events
(RX1day) occur, the quick runoff component exhibits minimal response during sub-periods 1 and 2. In contrast, the slow runoff process, $Q_s$, appears more influential, likely attributed to freeze-thaw cycles and snowmelt. For the subsequent sub-periods, $Q_s$, $Q_q$, and overland flow $OV$ all react to precipitation events. While $Q_q$ shows a slight decline, and the contribution of saturated runoff $OV$ increases. These changes improve the shortcomings of underestimating low flows and overestimating high flows in the runoff simulation process (Guo et al., 2018; Höge et al., 2018; Pande and Moayeri, 2018; Wang et al., 2018). (2) Experiment 7 provides
a more accurate simulation for diverse flow phases than experiment 1. (3) Anomalies in state variables occur in experiment 7 during sub-period 5, likely due to $H_{uz}$ constraints ($H_{uz} = 43$), maintaining $XH_{uz}$ and $XC_{uz}$ at low levels. This may arise from the optimization algorithm failing to converge, resulting in a local optimum 'trap' during this sub-period. However, experiment 7's simulation of flow in sub-period 5 is better than that of experiment 1, further demonstrating the robustness of experiment 7 and the reliability of dynamic parameter sets compared to an individual dynamic parameter. Similar results were observed in cases B, C, and
and D (see Fig. S5-S12). In sum, experiments 1 and 3 show negligible differences in state variables and flux time series. Experiment 4 fails to improve simulation results and triggers abnormal responses within certain model components. Experiment 5 exhibits invalid values, and experiment 6 encounters similar issues at sub-period boundaries during the validation period. Experiment 7 shows improved simulation performance, particularly in addressing the underestimation of high flows and overestimation of low flows, as evidenced by the model's internal variables.

The comparative analysis of experiment 1 and experiment 7 further illustrates the improvements in performance introduced by experiment 7. Fig. 5(c) and Fig. 5(d) illustrate the flux mapping of various sub-periods in study case A, comparing the control experiment 1 and the recommended scheme (experiment 7). Each scatter point in the figures represents a parameter set generated during the SCE-UA algorithm optimization process. The colour and relative position of each scatter point on the axes illustrate the variation in runoff components for sub-periods under specific parameter sets, as well as the corresponding objective function value.
To facilitate comparison, the results of experiment 1 are also presented by the same sub-periods as experiment 7.





Notably, the differences in optimization performance between experiments 1 and 7 reveal key insights into model behaviour. Both experiment 1 and experiment 7 show the poorest results in sub-periods 1 and 2, with the largest (worst) objective function values. In the remaining three sub-periods, the objective function values were significantly better. Compared to the traditional scheme, the recommended scheme consistently identified more optimal parameter sets with smaller objective function values within the same

period. Shifting the focus to flux components, the spatial distribution of scatter points in the flux maps reveals varied runoff components and internal model behaviour for each sub-period. Notably, despite similar objective function values, the recommended scheme possesses a narrower range of optimal equifinality parameters during the parameter evolution process, reducing the model's internal fluxes equifinality and uncertainty. From sub-periods 1 to 4, the SCE-UA algorithm more rapidly converges to near-optimal solutions, showing a narrower range of variability in the optimization process. In sum, the improvements

observed in experiment 7 not only highlight the importance of refining dynamic parameters but also underscore the model's capacity to simulate complex hydrological processes across different sub-periods.





**Figure 3.** Fluxes simulation results of experiments during the representative validation period for the case A. The figure shows the flux simulation results from Experiments 1 to 7, with different colours representing different sub-periods. In Experiment 7, five separate calibrations were performed for five sub-periods, and the results were then aggregated to obtain the final simulation.






**Figure 4.** State variables simulation results of experiments during the representative validation period for the case A. The figure shows the state variable simulation results from Experiments 1 to 7, with different colours representing different sub-periods. In Experiment 7, five separate calibrations were performed for five sub-periods, and the results were then aggregated to obtain the final simulation.



### 4.3 Parameters

The dynamic parameter sets, optimized by various calibration experiments across four case studies, are depicted in Fig. 5(b). Experiments 1, 2, and 3 utilized a time-invariant parameter set, adjusted through the objective function to reflect the catchment's average characteristics. Experiment 4 allowed the parameter $H_{uz}$, which exhibits the highest sensitivity, to vary across different sub-periods while maintaining other parameters constant. However, the dynamics of $H_{uz}$ in response to catchment characteristic changes across sub-periods did not significantly improve the model's performance within the four case studies. Experiment 5 allowed for all parameters to vary across sub-periods, but no consistent variation pattern emerged in response to catchment characteristics. In experiments 6 and the recommended experiment 7, certain parameters, such as $K_s$, exhibited minimal correlation with sub-period characteristics within catchments. $K_s$ value is highest during sub-periods with abundant and concentrated precipitation, higher temperatures, and higher antecedent runoff (soil moisture), and lowest during relatively cold and dry sub-period. However, this correlation was not significant. Furthermore, $K_q$ did not exhibit a clear pattern of variation, due to the poor response of α, which indirectly changed the model structure and bypassed the quick flow module. This phenomenon reflects the uncertainty inherent in model parameters and structure. In sum, compared to time-invariant schemes, the recommended scheme exhibits superior performance in identifying key parameters and their responses to catchment dynamics. The dynamic nature of parameters emphasizes the importance of calibration across sub-periods. Although the dynamic parameter set enhanced the model's response capability to seasonal dynamics, the overall response of the entire dynamic parameter set to seasonal dynamic characteristics remains relatively poor. The reasons for the improved simulation performance of the dynamic parameter set will be explored in the discussion section.



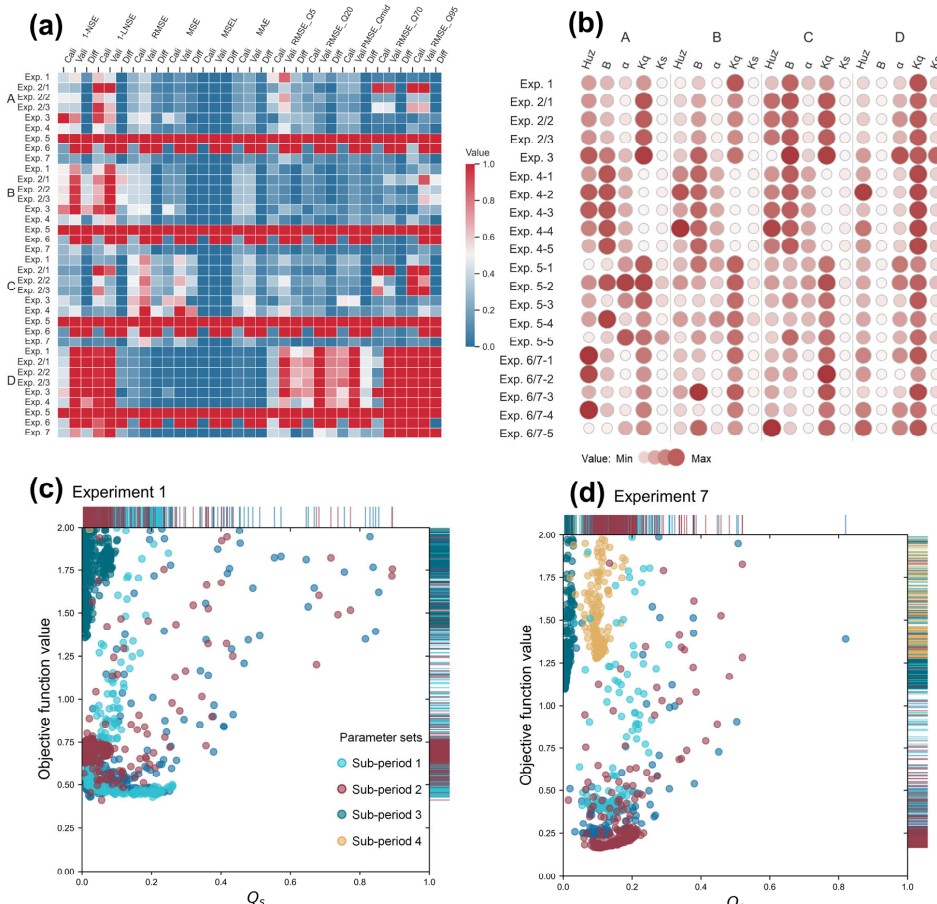

**Figure 5. (a)**. Model performance of the seven experiments in four study cases. **(b)**. Assessment of dynamic parameter sets across various calibration experiments. **(c)/(d)**. Flux mapping for case A in both the conventional **(c)** and recommended **(d)** schemes, where the horizontal axis represents the proportion of $Q_s$ in the runoff.

## 5 Discussion

### 5.1 Why dynamic parameter sets improve simulation performance

Despite the significant improvement in the simulation performance of hydrological models based on seasonal dynamic characteristics, the response of discretized dynamic parameters (even highly sensitive ones) to these seasonal dynamics is not satisfactory. However, a dynamic parameter set can collectively carry the extracted information of seasonal dynamic features, compensating for model structural deficiencies and improving model performance. Therefore, this study further explores the potential reasons from three aspects: the correlations between parameters, equifinality in the hydrological model, and the evolution process of parameters.



### 5.1.1 Complex correlation between parameters

Fig. 6(a) and Fig. 6(c) demonstrate that there are both significantly linear and nonlinear correlations among the parameters of the hydrological model in study case A (results of other cases are shown in supporting information). MIC values above 0.35 among most parameters suggest that the dynamics of individual parameters may be affected by others (Bárdossy, 2007). This explains unimproved model performance when altering individual parameters during different sub-periods in experiment 4. The analysis results of parameter sensitivity based on scatter plot method also confirmed the influence of the correlation between parameters. In the recommended scheme (experiment 7), parameter like $K_s$ (the slow-flow routing tank's rate) exhibit a weak responsive relationship to the seasonal dynamic of the catchment, validating the significance of clustering sub-periods based on catchment dynamics. Due to the complex linear or nonlinear correlations between parameters, the variation of individual parameters can be compensated for by changes or adjustments in other parameters, leading to no significant changes in the simulation performance of the model (Xiong et al., 2019a; Vaze et al., 2010; Zhou et al., 2022). Bárdossy (2007) suggested that parameters within a hydrological model parameter group should not be considered individually but rather treated as a whole.

### 5.1.2 Equifinality in the hydrological model

The parameter sets derived by the SCE-UA algorithm for flux mapping encounter inherent limitations (Beven, 1993; Padiyedath Gopalan et al., 2018). This arises due to the algorithm's inherent directionality in the optimization process, which potentially overlooks certain parameter sets capable of producing equifinality results. Analysis of parameter sensitivity through flux mapping and scatterplot methodology reveals a distinctive feature towards the end of the search path: a tail-like pattern in the scatterplot in Fig. 5(c) and Fig. 5(d), indicating a series of parameter sets with equifinality identified by the optimization algorithm. These scatter points represent parameter sets producing similar results, though originating from distinctly different physical processes. Hence, it may fail to infer that model runs exhibiting higher performance values consistently correspond to more realistic scenarios. The evaluation of model performance, particularly when quantified in a scalar manner, emerges as a weak, unreliable, and unrealistic approach for model assessment. The representation of model processes cannot be sufficiently measured by a solitary performance metric or a limited range of values (Khatami et al., 2019; Gupta et al., 2009a; Santos et al., 2018). A rigid interpretation of objective functions can lead to misinterpretations; for instance, in Fig. 6(b) model runs with marginally lower NSE values might offer more realistic underlying processes compared to those with better NSE values (Gomez, 2019). It is vital to acknowledge that high model performance does not inherently equal realism and may be influenced by numerical artifacts arising from various sources of uncertainty. Moreover, our constrained understanding of catchment processes, involving runoff generation mechanisms and complex runoff events, makes it challenging to determine the likelihood of specific parameter sets occurring in reality (Clark et al., 2015).

### 5.1.3 Evolution process of parameters

While the causes of abnormal dynamic parameter values are complex, they might be partially attributed to the failure of global optimization algorithms to converge and find approximated global optimal solutions during the evolutionary process. Hydrological model parameter response surfaces exhibit a range of complex characteristics, including high non-linearity, multi-modality, non-convexity, irregularity, discontinuity, noise, roughness, and non-differentiability (Weise, 2009; Maier et al., 2014; Kallel et al., 1998). To better describe the evolutionary process of the parameters, a fitness landscape is used, where the vertical axis represents the objective function values and the horizontal axis represents the parameter space (Fig. 6(d)). The evolutionary process is the process of searching for a global optimum. During this process, deceptive gradients of the objective function values can mislead





the optimizer away from the global optimum; the increase in the number of local optima also makes the search path for the global optimum more complex and challenging (Weise, 2009). Terminating at a local optimum can prevent the optimized parameters from accurately responding to environmental changes.

**5.2 Problems caused by parameter abrupt shifts**

Abrupt parameter shifts disrupt the assumption of long-term water balance in traditional hydrological models, potentially leading to invalid values for state variables in adjacent sub-periods. (Kim and Han, 2016; Beven and Binley, 2006; Sivakumar, 2004; Laloy and Vrugt, 2012). For instance, during the transition of soil maximum storage height ($H_{uz}$), the $H_{uz}$ value for the next sub-period might be lower than the former actual state variable value ($XH_{uz}$). Similarly, numerical overflow errors might lead to model

crashes and the generation of invalid results (Fig. 6(e)). These errors could also propagate through various modules of the model, such as the high-speed runoff module and slow-speed runoff module, disrupting the proper functioning of other parts of the model, and causing the optimization algorithm incapable of producing valid results.

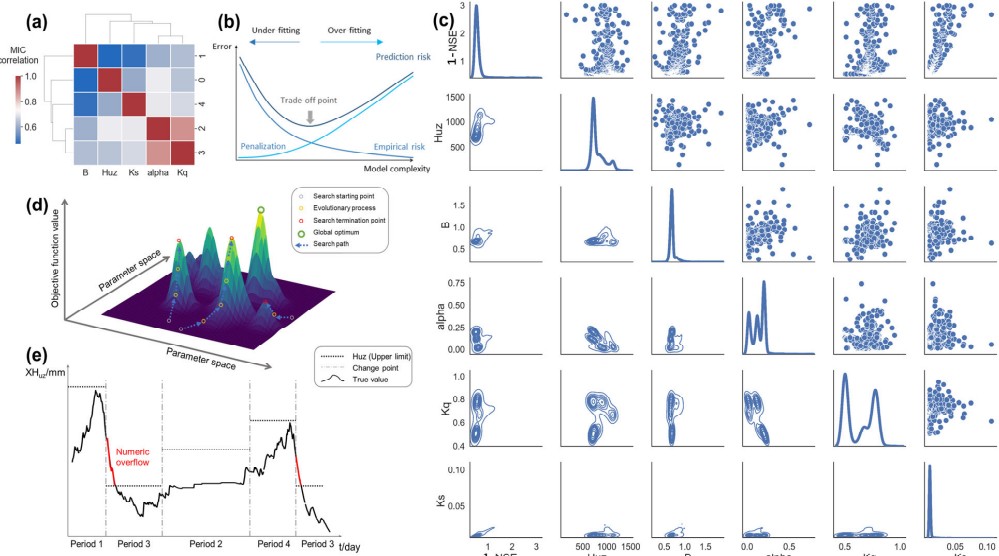

**Figure 6. (a)**. Linear or nonlinear correlations between parameters based on MICs in case A, with red indicating the strongest correlation among

parameters. **(b)**. Conceptual diagram illustrating the trade-off between empirical fitting to data and the penalization of model complexity, and its impact on prediction error (Schoups et al., 2008). **(c)**. Parameter sensitivity analysis for case A through scatter plots. **(d)**. Three-dimensional fitness landscape showing the objective function values on the vertical axis, parameter space on the horizontal axis, and various evolutionary paths that elements can follow within the parameter space, indicated by arrows. **(e)**. Conceptual diagram of errors resulting from abrupt parameter shifts.

**6 Conclusions**

Due to limitations in observational data and a lack of understanding of catchment hydrological processes, the structure of traditional conceptual hydrological models often fails to represent catchment dynamics, leading to inaccurate model simulations. Here we



aimed to develop a methodology to effectively calibrate the hydrological models, addressing the limitations imposed by model structure. The Extracting Seasonal Dynamic Catchment Characteristics (EDCC) approach was taken to cluster the dynamic

processes of the catchment into diverse sub-periods, linking hydro-meteorological data with models through objective function and dynamic parameters. The identified clusters were calibrated using seven contrasting calibration experiments applied to the MOPEX dataset and comprehensively assessed from the perspectives of internal model processes. A recommended calibration scheme was ultimately proposed. The following specific conclusions could be drawn from this study:

- Adjusting the configuration of the objective function can enhance the simulation of emphasized flow phases, but at the cost

of sacrificing simulation performance for other flow phases, making it difficult to improve overall model performance.

- Due to issues of model structural deficiencies, correlation among parameters, dimensional disaster in optimization, and the transition of dynamic parameters between adjacent sub-periods, improving model performance through individual parameters alone is not feasible. Model parameters should be considered as a group of parameters.

- Based on the EDCC approach, which clusters sub-periods according to catchment dynamics, the recommended scheme

achieved better simulation results. Sub-period calibration enables conceptual hydrological models to overcome structural deficiencies and simulate the dynamic processes of the catchment.

The calibration and evaluation framework proposed in this study not only addresses defects caused by the simplification of model structure for hydrological models but also enhances model simulation accuracy across different flow phases and effectively reduces model uncertainty. The evaluation framework comprehensively assesses the performance of hydrological models through multi-

criteria evaluation and reveals sources of uncertainty in model internal operation from the perspectives of state variables and fluxes. Despite the positive results of this study, developing more realistic models will aid in our understanding of hydrological processes and improve hydrological forecasting.

**Supplement link**

Supporting Information are provided.

**Author contributions**

Tian Lan devised the modelling concept. Tian Lan and Xiao Wang wrote the code, and prepared the original draft manuscript. Hongbo Zhang, Xinghui Gong, Xue Xie, Yongqin David Chen and Chong-Yu Xu provided supervision, and reviewed/edited the manuscript.

**Code availability**

The MOPEX dataset is available at Duan et al. (2006). The Sensitivity Analysis For Everyone (SAFE) toolbox is available at https://safetoolbox.github.io/ (last access: 23 November 2024) (Pianosi et al., 2015). Model set up configurations have been reported in https://zenodo.org/records/11058234 (last access: 23 November 2024).





**Competing interests**

The authors declare that they have no conflict of interest.

**Acknowledgments**

This study is financially supported by the the National Natural Science Foundation of China (NSFC) (Grant No. 52209006, 52379003, and 42071055), China Postdoctoral Science Foundation (Grant No. 2021M700018), and Research Council of Norway FRINATEK Project 274310.

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
