# Peer review of "A Novel Framework for Calibration and Evaluation of Hydrological Models in Dynamic Catchments"

_Hydrology and Earth System Sciences, 2024_

## Referee Comment (RC1)

**Summary**

The manuscript presents a proposed calibration framework for hydrological models in dynamic catchments, centred around the integration of dynamic catchment characteristics and sub-period-based calibration using time-varying parameters. While the overall goal of improving hydrological model performance in non-stationary systems is both relevant and timely, I found the manuscript difficult to follow and insufficiently developed in terms of methodological clarity, result presentation, and scientific argumentation.

The potential scientific contribution—particularly the EDCC method and its integration into a calibration framework—is undermined by major issues in structure, presentation, and depth of analysis. These issues make it difficult to properly evaluate the scientific quality and significance of the work.

**Major Comments**

**Clarity and Structure**

The manuscript's presentation is a major limitation to its scientific communication. While clarity should not override substance in peer review, in this case the lack of structure and clarity severely affects the reader's ability to assess the methods and results.

- The **"Methods"** section is disorganized and lacking in detail. For instance:
  - Section 2 on *Dynamic catchment characteristics* should be integrated into the Methods, as it is a central component of the proposed framework.
  - The EDCC approach is only superficially described in the main text, despite its centrality to the framework's novelty. Key implementation and performance results are hidden in the SI, where they are difficult to evaluate.
  - Details on the hydrological model (HYMOD) are insufficient. Given the extensive use of model fluxes and state variables in both analysis and figures, a clearer introduction to the model and its components is essential.
  - The calibration experiments, though commendably framed with clear objectives, lack clarity in terms of execution and consistency:  It is essential to clarify what is being optimized, when, and how parameters are treated during sub-periods. For example, lines 146-155 reference SCE-UA, while Experiment 2 introduces NSGA-II without clearly stating if other experiments revert to SCE-UA. The description of Experiment 4, in particular, remains opaque even after repeated readings. Figure 2 is helpful but insufficient.
  - In general, the reader should not need to reference the SI repeatedly to understand the core methodology.
  - With respect to the "Evaluation" section, more detail should be provided on each performance metric, including references and benchmark values. Also note that Table 1 and Table S2 are almost identical and redundant.

- The discussion of flux mapping is vague; although described as a ternary plot method, such plots do not appear in the main text.

- In the **"Results"** section, the presentation is often superficial and uneven.

  - Although four case studies are introduced, results are shown primarily for Case A. In most instances, other cases are summarised with a single sentence asserting similarity. If the intention is to generalise the proposed framework, this is inadequate. Either more contrast between cases should be shown or a more compelling rationale for their selection should be provided.

  - A comprehensive synthesis across the full set of catchments (e.g., the 130 dynamic basins in MOPEX) is conspicuously missing. Only a few lines (291–297) address this aggregation, and no discussion is offered on spatial or climatic variability in model performance.

  - EDCC results should be presented within the Results section, not just described or relegated to Supporting Information.

  - The analysis of parameter correlation and flux mapping—currently discussed in the Discussion—should be integrated as part of the core results. These are not interpretive reflections, but rather diagnostic outputs central to evaluating the model framework.

These limitations and in the structure of the manuscript are compounded by the choice of the figures, which, while informative in parts, suffer from poor organisation and mixed messaging:

- Figures such as Figure 1 and Figure 6 combine multiple purposes (contextual information, results, conceptual illustrations) in a way that muddles their message. Each figure should ideally present a single, focused point.

- Visuals that directly illustrate performance improvements are lacking. Figure S4 (which compares calibration outcomes across all basins) should be elevated to the main text.

- Conversely, time series plots (Figures 3 and 4) do not meaningfully add to the manuscript and could be moved to the SI if needed.

**Scientific Significance and Depth**

Despite its potential, the scientific contribution of this work is undermined by superficial analysis and poor framing of the results:

- The conclusions (lines 444–451) include trivial points (e.g., trade-offs in multi-objective calibration) that do not substantively add to the field. The third point—regarding sub-period calibration as a remedy to structural model deficiencies—is more interesting, but is not adequately supported by clear, main-text results.

- The claim that this is a "novel framework" requires broader evidence of generalisability and applicability across a wide range of catchments. How does performance vary with catchment type, climate regime, or data quality?

- The "**Discussion"** section falls short of its purpose. It lacks depth, avoids key limitations, and does not engage with broader literature on parameter identifiability or structural

uncertainty. In particular, there is a missed opportunity to discuss important limitations and implications. For example:

- What are the limitations or assumptions of the EDCC clustering approach?
- How does the framework handle equifinality and model realism, beyond scalar performance metrics?
- Why do dynamic parameters fail to reflect environmental signals in some experiments (e.g., Experiment 4)? Are the algorithms or model structures to blame?
- How are the results influenced by the specific structure of the HYMOD model? How generaliseble are they?

**Conclusion and Recommendation**

While the topic is of significant relevance to the hydrology community, the manuscript in its current form suffers from major shortcomings in clarity, structure, and depth of analysis. The central innovation (dynamic sub-period calibration) is potentially valuable, but is not convincingly demonstrated or critically discussed. The supporting methods (e.g., EDCC) and results are insufficiently explained or buried in supplementary materials, making it difficult to assess the true scientific merit.

I recommend that the authors reconsider the scope and objectives of the manuscript and develop a substantially revised version that clearly communicates the methodology, demonstrates the performance improvements across diverse settings, and meaningfully engages with the implications and limitations of the proposed approach.

Recommendation: Reject, but encourage resubmission after significant restructuring.

**Minor Comments**

- Line 1 (title): I'm not sure if the novel approach relates to evaluation, also I have never heard of "dynamic" catchments, maybe seasonal is a better term here.
- Line 35: projecting, rather than predicting is a better term in this contextual
- Line 85 (and throughout): the term "dimensionality disaster" sounds very grandiose and pompous, I would avoid it, but needs to at best be better defined.
- Line 90: experiments are conducted, not verified.
- Lines 99-101: the criteria used here should be better explained (even in the SI). In particular criteria 2 and 3 feel very subjective.
- Lines 111-123: this section requires better referencing of the methods described
- Lines 114, 117, 119: check the format of the bullet points here.

- Lines 132-133: "calibrating" and see point on line 1 with respect to "dynamic catchments"

- Line 133: the Pareto-based method need better introduction and references.

- Line 141: despite their name, in a simple conceptual model, parameters don't really have "clear physical meaning".

- Line 142: the reference to Fig 2 here seems out of place, please check

- Line 152 (and throughout): "validation" is a rather controversial term when it comes to modelling. Please use "evaluation" instead.

- Lines 152-154: the sentence "It should be noted…", while true feels deceptive. The authors did not actually test this with any other model.

- Line 181: be careful with the use of acronyms, and do not introduce acronyms that haven't been explained previously.

- Line 186: "Parameters" … "are", or "The parameter" … "is".

- Line 205: The parenthesis "(n is the number of sub-periods)" is redundant

- Line 212: In this example, n is five, this sentence needs rewriting.

- Line 255: "Flux mapping"

- Line 291: What "evaluation metrics" are being referred to here? Is this an average of all of them? There needs to be additional clarity on how this is being evaluated.

- Line 343 and 348 (Figures 3 and 4): Are these calibration or evaluation results? The caption says one thing, the legend a different one.

- Line 410: "abnormal" seems like a charged word for this